# A Map of 3′ DNA Transduction Variants Mediated by Non-LTR Retroelements on 3202 Human Genomes

**DOI:** 10.3390/biology11071032

**Published:** 2022-07-08

**Authors:** Reza Halabian, Wojciech Makałowski

**Affiliations:** Institute of Bioinformatics, Faculty of Medicine, University of Münster, 48149 Münster, Germany; reza.halabian@uni-muenster.de

**Keywords:** DNA transduction, structural variants, mobile element insertions, human genomics, retrotransposon, 1000 Genomes Project

## Abstract

**Simple Summary:**

During the transcription of non-LTR retroelements, such as LINEs and SVAs, the transcriptional termination signal at the 3′ end might be ignored by RNA polymerase. As a result, the transcription terminates at another downstream signal, creating a chimeric transcriptional readthrough. Termed 3′ DNA transduction, this process copies the 3′ flanking region along with the retroelement sequence to a new genomic locus, which influences the structure of the genome and occasionally possesses a functional impact. To discover putative non-LTR retroelement-driven 3′ DNA transductions, we analyzed the new dataset (n = 3202) of the 1000 Genomes Project. Our results indicate that 3′ transduction derived by non-LTR retroelements is a relatively common phenomenon in the human genome and that their discovery needs to be more appreciated in genome projects.

**Abstract:**

As one of the major structural constituents, mobile elements comprise more than half of the human genome, among which *Alu*, L1, and SVA elements are still active and continue to generate new offspring. One of the major characteristics of L1 and SVA elements is their ability to co-mobilize adjacent downstream sequences to new loci in a process called 3′ DNA transduction. Transductions influence the structure and content of the genome in different ways, such as increasing genome variation, exon shuffling, and gene duplication. Moreover, given their mutagenicity capability, 3′ transductions are often involved in tumorigenesis or in the development of some diseases. In this study, we analyzed 3202 genomes sequenced at high coverage by the New York Genome Center to catalog and characterize putative 3′ transduced segments mediated by L1s and SVAs. Here, we present a genome-wide map of inter/intrachromosomal 3′ transduction variants, including their genomic and functional location, length, progenitor location, and allelic frequency across 26 populations. In total, we identified 7103 polymorphic L1s and 3040 polymorphic SVAs. Of these, 268 and 162 variants were annotated as high-confidence L1 and SVA 3′ transductions, respectively, with lengths that ranged from 7 to 997 nucleotides. We found specific loci within chromosomes X, 6, 7, and 6_GL000253v2_alt as master L1s and SVAs that had yielded more transductions, among others. Together, our results demonstrate the dynamic nature of transduction events within the genome and among individuals and their contribution to the structural variations of the human genome.

## 1. Introduction

Annotation of the telomere-to-telomere human genome indicates that more than half of our genome is populated with repetitive sequences, including mobile elements (MEs), also known as transposons or transposable elements (TE) [1]. Of these TEs, *Alu*s, L1s (LINE-1), and SVAs—all classified as non-LTR retrotransposons—are still active and continue propagating within the human genome [2]. A common feature of these active elements, particularly L1s and SVAs, is a process called 3′ DNA transduction, by which retroelements co-mobilize DNA flanking their 3′ end to new genomic loci [3]. L1/SVA-driven transduction results from a weak 3′ polyadenylation signal in the source element, which might be skipped by RNA polymerase II during transcription. In this scenario, transcription continues until another downstream polyadenylation motif is reached [4]. Several studies have shown that DNA transductions mediated by retroelements are a common biological phenomenon in the *Homo sapiens* genome [3,5,6,7,8,9,10]. For instance, Hoyt et al. estimated that the transduction rate in the T2T-CHM13 genome (the complete human genome) was 0.026 events per 1 Mbp [1].

Mobile elements can generally give rise to genome instability by insertion, mutation, and rearrangement [11]. They also influence the transcriptome via mechanisms such as alternative splicing and can alter the epigenome by the generation of differentially methylated regions (DMR) [12]. DNA transductions mediated by mobile elements can also affect the genome in several ways, from shuffling genomic DNAs to duplicating genes [3,4,5,7]. Sequence shuffling can involve exons or regulatory elements such as enhancers and promoters, creating new regulatory region combinatorics [4]. Moreover, it has been demonstrated that the transduction process is involved in developing some diseases and is a possible source of somatic mutations in the tumorigenesis [13]. For instance, in a study across 290 samples, Tubio et al. reported extensive 3′ transductions in lung and colorectal cancer patients [9].

Thanks to the rapid development of sequencing technologies, several national and international genome projects, such as the 1000 Genome Project (1KGP), have been launched to comprehensively characterize different types of genomic variants (i.e., single nucleotide variant (SNVs), short insertions/deletions (indels), and structural variants (SVs)), their origin, and their effect on human populations [14,15,16,17,18,19]. MEIs (Mobile Element Insertions) are responsible for almost 25% of SVs in the human genome [20], and consequently, they are an important driver of SVs [21]. At the population scale, active MEs are polymorphic concerning their presence or absence across individuals [22]; thus, they can also potentially produce transduction variants within/among populations. Although several studies have cataloged polymorphic L1s and SVAs in the 1KGP dataset [20,22,23,24], a few have investigated transductions [20,25,26]. To our best knowledge, with one exception [10], all these studies focused on the transductions mediated by L1s based on the low-coverage dataset of 1KGP. 

It has been estimated that the new insertion rate is about 0.5–1 and 0.1 per 100 live human births for L1s and SVAs, respectively [27,28]. This implies that many MEIs are absent in the reference genome assembly but can be revealed by analyzing different populations. As a consequence of such dynamic propagation, the detection of transduction events is expected to increase proportionally as well. Recently, the New York Genome Center (NYGC) has released the whole-genome sequence data for 3202 individuals at high coverage (30X), hereafter referred to as the new panel of the 1KGP, in which 698 samples are new individuals added to the previous dataset [29]. The new dataset benefits from a higher sequencing coverage and a longer read length. Hence, it can considerably improve variant discovery power, delivering more precise breakpoints, mapping resolution, and genotype accuracy. This can be particularly lucrative for identifying MEIs and transduction events, given the repetitive nature of these sequences. 

In this study, we applied a set of computational approaches to probe and profile putative 3′ transduction variants mediated by non-LTR retroelements in the new panel of the 1KGP. Here, we show that 3′ transduction is a dynamic and relatively common phenomenon in the human genome. We also provide a comprehensive genome-wide map and annotation of transduced segments, including their chromosomal location, length, allelic frequency, progenitors, and distribution pattern within populations. Additionally, we present the most productive retroelements (master elements and their loci). Given the lack of a broad investigation of non-LTR retroelements-driven transductions in the new panel of the 1KGP, we hope the results of this research can serve the scientific community for a better understanding of the biology, evolution, and structure of the human genome.

## 2. Materials and Methods

### 2.1. Data 

To identify 3′ transductions mediated by L1s and SVAs, we utilized the new panel of the 1KGP. Collectively, 3202 high-coverage (~30X) whole-genomes data mapped to the GRCh38 human reference genome (GRCh38_full_analysis_set_plus_decoy_hla.fa) were downloaded (CRAM format, accessed on 5 August 2020) from the European Nucleotide Archive (ENA) (ftp://ftp.sra.ebi.ac.uk/vol1/run/, accessed on 5 August 2020) [30,31]. The new panel of the 1KGP consists of 26 populations organized into five distinct super-populations; African, Ad Mixed American, European, South Asian, and East Asian (Table 1).

### 2.2. Discovery of MEI Polymorphisms

First, all samples were indexed using the following command: *samtools index <genome.cram>* [32]. In order to identify and characterize 3′ transduction events, we applied the Mobile Element Locator Tool (MELT v2.2.0), which was developed as a part of the 1KGP [20]. MELT is a software package and a multi-step program; hence, discovering transduction events requires running different commands and processes (Figure 1). Since identifying polymorphic non-LTR retroelements was a prerequisite for transduction discovery, we utilized MELT in the SPLIT mode to find polymorphic non-reference MEIs (the length of the consensus L1 and SVA were 6019 and 1316 nucleotides, respectively). For this purpose, we implemented the Preprocess, IndivAnalysis, GroupAnalysis, Genotype, and MakeVCF modules, with the default parameters except for the followings: [i] in the IndivAnalysis module, -r was set to 150 (read length) and [ii] for -v in the GroupAnalysis module, which specifies the discovered variants in the previous studies, the prior VCF file was downloaded from NCBI ftp server (https://ftp.ncbi.nlm.nih.gov/pub/dbVar/data/Homo_sapiens/by_study/vcf/nstd144.GRCh38.variant_call.vcf.gz, accessed on 10 August 2020). Finally, for generating the final VCF files required for transduction discovery, we used BCFtools [32] to filter out the low confidence polymorphic MEIs identified across 3202 genomes; therefore, we kept those variants tagged as “PASS” or “rSD” in the FILTER column of the corresponding VCFs as well as their ASSESS (the score related to available evidence on the breakpoint of insertion) and SR values (total number of split reads at the insertion location), which were ≥three:
*bcftools view -i ‘SR >= 3* && *ASSESS >= 3′ <input.vcf> | bcftools view -i ‘FILTER=“PASS” || FILTER=“rSD”‘ | bcftools view -e ‘FILTER=“ac0” || FILTER=“hDP” || FILTER=“lc”‘ -o <output.filtered.vcf>*

where rSD indicates that all potential variants should be included in the results despite how balanced is the support on the left and right side of the breakpoint. All the potential variants for which support for the left and the right side of the breakpoint differ by two standard deviations are labeled rSD. By keeping these variants, we increase the transduction discovery sensitivity. 

**Figure 1 biology-11-01032-f001:**
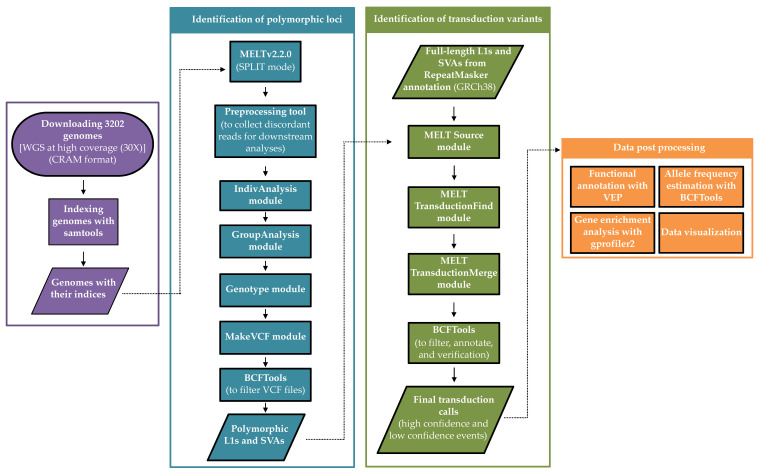
Schematic overview of the applied workflow to identify and characterize 3′ DNA transductions mediated by non-LTR retroelements.

### 2.3. Identification of 3′ Transduction Events

We employed the transduction tool from MELT to identify 3′ transduction events among populations. In general, this tool investigates 3′ ends of reference (annotated by the RepeatMasker) and non-reference non-LTR retroelements (present in the VCFs) to find discordant read pairs that map to another site right adjacent to a known non-reference MEI [20]. This analysis was performed by implementing three modules respectively: Source, TransductionFind, and TransductionMerge. The Source module was fed by the VCF files obtained from the previous step (2.2). Additionally, a BED file containing possible source elements (L1s and SVAs) within the reference genome was required for the Source module. We prepared these BED files for each element by downloading the RepeatMasker track from the UCSC genome browser for GRCh38 (http://genome.ucsc.edu/cgi-bin/hgTables) (accessed on 2 June 2021). It has been shown that a considerable number of L1s and SVAs are 5′ truncated [3,33], which disrupts their amplification and transduction potential. Thus, to avoid any ambiguity, we only included the full-length L1s and SVAs in the analysis; the minimum length was set to 5900 and 1000 nucleotides for L1 and SVA elements, respectively, for the Source module. The other modules of the transduction tool were implemented according to the MELT manual (https://melt.igs.umaryland.edu/manual.php, accessed on 2 June 2021). Finally, a non-LTR retroelement was regarded to carry a transduced sequence if their “METRANS” field was not annotated as “null” in the resulting VCF files. 

### 2.4. Data Post-Processing (Validation, Classification, and Allele Frequency Estimation)

To generate the final lists of transductions, we first removed potential transductions overlapping with segmental duplication loci (downloaded from UCSC table browser, https://genome.ucsc.edu/cgi-bin/hgTables, accessed on 1 September 2021). We then defined a two-tier validation scheme, based on which we categorized the preliminary results into high- and low-confidence transductions. A transduction event was classified as a high-confidence event if five or more reads supported it and the reported length of the transduced segment at the source was shorter than reported in VCF file length of the offspring, otherwise classified as a low-confidence event. The latter option is a consequence of MELT-specific VCF file. The transduction length is reported there as the whole structural variant, i.e., TE length + traduced element length and is stored in the column named SVLEN. However, the progenitor length is reported only based on the length of the segment that was transduced without the length of the TE and is stored in the column named MESOURCE [length]. Consequently, we should expect the offspring length (SVLEN) to be longer than MESOURCE [length] by more or less the size of a TE (5 kb in the case L1 events and 1 kb in the case of SVA events). Additionally, we checked the family type of all source elements that gave rise to the identified transductions. Consequently, if they did not belong to an active retroelement (Appendix A) in the recent human genome, we classified them as low-confidence events. Finally, for each high-confidence transduction variant, the allele frequency was estimated using bcftools as follows:
*bcftools plugin fill-tags <input.transductions.vcf> -o <output.transductions.AF.vcf > -- -S sample.txt -t AF,AC*


### 2.5. Functional Annotation

In order to find the insertional location and annotate the potential functional impact of each transduction variant on transcripts and regulatory regions, we applied VEP, the Ensembl Variant Effect Predictor v103.0 [34], as follows:
*vep --cache --distance 2000,1000 --regulatory --numbers --variant_class --overlaps --gene_phenotype --canonical --symbol --tab --terms SO --pick -i <tab-delimited format> -o <output.tsv>*

### 2.6. Functional Enrichment Analysis

To investigate potential common features present in the list of affected genes by retroelements carrying transductions, we used the gost tool from gprofiler2 (R interface) [35,36] to search eleven functional biological repositories (GO molecular function, GO biological process, GO cellular component, KEGG [Kyoto Encyclopedia of Genes and Genomes], Reactome database, WikiPathways, Human Protein Atlas, CORUM database, TRANSFAC, miRTarBase, Human phenotype ontology), as follows:
*functional_analysis <- gost(<gene_list>, organism = “hsapiens”, ordered_query = FALSE,multi_query = FALSE, significant =TRUE, exclude_iea = TRUE, correction_method = “gSCS”, domain_scope = “annotated”, user_threshold = 0.01)*


In this regard, our gene list consisted of 290 Ensemble IDs including UTRs, regulatory, intronic, coding, noncoding, TFBS, and the upstream of genes variants (as annotated by VEP). Finally, the results were visualized using the following commands:
*gostplot(functional_analysis, capped = TRUE, interactive = FALSE)**publish_gosttable(functional_analysis, highlight_terms = functional_analysis$result[c(1:3,7,9, 4:6, 8, 10),], use_colors = TRUE,show_columns = c("source", "term_name", "term_size", "intersection_size"),filename = NULL)*

### 2.7. Data Visualization

To visualize the size of transductions, the shared loci among populations, and the chromosomal distribution of transductions, we employed R packages: ggplot2 [37], VennDiagram [38], UpSetR [39], and RIdeogram [40]. In order to depict the genome-wide allele frequencies and progenitor-offspring relationships, we used the Circos tool [41]. 

## 3. Results and Discussion

### 3.1. Polymorphic Non-Reference MEIs

As a prerequisite step for transduction discovery by MELT, we identified polymorphic non-reference retroelements across 26 populations. In total, we detected 7103 and 3040 polymorphic loci for L1 and SVA elements, respectively (Table 2, VCFs S1 and S2). Our results show that the number of identified polymorphic (L1 and SVA) loci is the highest in the AFR super-population, whereas the lowest number was observed in the EUR one (Figure 2A). We found that the mean of L1 and SVA insertions per individual was 144.3 and 52.6, respectively. Not surprisingly, these numbers are smaller for the low-coverage dataset of the 1KGP (L1: 119 and SVA: 28.6 [26]), as lower coverage data tend to result in a high rate of false negatives. This clearly demonstrates the impact of higher sequencing depth on variant detection. Looking at the genome-wide density of polymorphic non-reference L1s and SVAs (Figure 2B), we observed that chromosome Y owned the lowest insertional rate with 0.052 insertions per 1 Mbp compared to chromosome X (2.94 insertions per 1 Mbp) and other autosomes (3.26 insertions per 1 Mbp on average). The same trend was also detected when polymorphic *Alu* insertions were included in the calculation of insertional rate; 0.21, 9.02, and 13.2 per 1 Mbp for the Y chromosome, X chromosome, and other autosomes, respectively (results are not shown). From a technical point of view, this might be partly related to the fact that GRCh38-Y has the highest fraction of unknown sequences (58.69% including hard masked PARs, Pseudo Autosomal Regions) among the other chromosomes, prompting it impossible to detect MEIs within these regions. On the other hand, given the highly repetitive nature of chromosome Y [42], MELT excludes reads mapped to multiple locations from the MEI discovery pipeline. Consequently, many regions in the Y chromosome are not available for analysis. This can also explain the low rate of identified polymorphic sites within this chromosome. Taking all these considerations together, MEI detection on the Y chromosome seems to remain a challenge, especially using short reads. 

Investigating the genotypes of each insertion among individuals, we found SVAs to be more heterozygous than L1s; the proportion of heterozygous to alternative allele homozygous, per-site basis, was 5.64 and 13.68 for L1s and SVAs, respectively. To further explore the contribution of genotyping errors to the observed heterozygosity, we performed the Hardy–Weinberg Equilibrium (HWE) analysis using an exact test [43] for each insertion. Although 9.61% of L1s and 5.49% of SVAs showed deviation (*p* < 0.01) from HWE, only 2.37% (169 loci) of L1s and 2.86% (87 loci) of SVAs violated (*p* < 0.01) the HWE excess heterozygosity, which signified the accuracy of genotypes [44,45]. We found that many polymorphic loci had an allele frequency of less than one percent, rare variants, across five super-populations (Figure 3A). This observation is consistent with Niu et al.’s results [26]. According to Niu et al. [26], the sample size is apparently a factor in the number of detected MEIs and thus observed allele frequencies; they demonstrated that these numbers raised with the increasing sample size [26]. However, it has been known that polymorphic MEI loci generally tend to be present at low allele frequencies [24]. 

Compared to Byrska-Bishop et al.’s results [29], in which they used the same dataset (the new panel of the 1KGP), we detected more polymorphic MEIs (Figure 3B). This difference in the number of identified polymorphic MEIs might pertain to their study’s pipeline and conservative filtration. Briefly, they discovered and assessed MEIs using GATK-SV [46]; MELT was the core algorithm, but was followed by a random forest technique to filter the preliminary variants conservatively [47]. We, nevertheless, were able to recover 74.79% of Byrska-Bishop et al.’s calls (L1s and SVAs) with the exact allele match in our dataset. In concordance with this, we also identified more polymorphic MEIs than Gardner et al. [20]. Interestingly, Gardner et al. [20], who used the low-coverage data from the 1KGP for their analyses, discovered more polymorphic L1s and SVAs than Byrska-Bishop et al. These discrepancies partly imply that MEI discovery is not an easy task, specifically using short reads. Therefore, combining different heuristic approaches to capture more true variants appears to be necessary. 

### 3.2. Transduction Variants

#### 3.2.1. Number of Identified 3′ Transductions

In order to identify 3′ transductions mediated by L1 and SVA elements in the new panel of the 1KGP, we analyzed 3202 genomes. We applied several criteria to validate the identified transductions. First, we discarded the transductions located in the segmental duplication sites due to the confounding effects of these regions on the transduction signatures that complicate the accurate inference of the progenitor–offspring relationship in a transduction analysis [6]. Then, the remaining candidates were classified as high- or low- confidence transductions (see methods 2.4 for more details). In total, we identified 268 high-confidence 3′ transductions mediated by L1s and 162 SVA transductions (see Table 3 and Appendix A). 

As of writing this manuscript, there was no retroelement-driven transduction investigation on the new panel of the 1KGP. However, a few studies on L1s using the low-coverage data from the 1KGP were carried out, in which, except one case [26], the older versions of the human genome assembly have been used as a reference genome [20,25]. Moreover, the investigation of SVA-driven transductions in this study seems to be the first work on the all samples of 1KGP (both the old and new panel). The comparison of our findings shows fewer identified L1 transductions (268 high-confidence events from 63 progenitors) (Appendix A) than Niu et al.’s work (457 events from 58 donor loci), in which they used the low-coverage genomes [26]. This difference is most likely due to the filtration scheme we have applied. For instance, in contrast with Niu et al., we excluded transduction within segmental duplication areas. Surveying Gardner et al.’s work [20], in which the hg37 genome build was used, we found more L1 transduction events in our study (268 vs. 121) and we were able to recover 30 (out of 121) L1 transductions plus one L1 hotspot (chromosome 6: 13190802) in our results (Appendix A). In this regard, similar discrepancies between the findings of Gardner et al. and Niu et al. (both have used the same dataset, i.e., low-coverage samples) have been reported [26]. In addition to the applied filtration and classification criteria in our analysis, other factors that stemmed from the 1KGP dataset, including differences between the read length (low-coverage: 76–101 bp, high-coverage: 150 bp) and sequencing coverage (low-coverage: mean depth ~7.4X, high-coverage: mean depth ~34X) [29], more likely have contributed to the observed differences between our and previous studies. In a study to produce diploid haplotype-resolved assemblies and identify the whole spectrum of structural variants within a small sub-set of 1KGP samples (n = 35), Ebert et al. detected 32 3′ SVA transductions [10]. We were able to find only one overlap from their list with our calls (Appendix A). In this case, there was no evidence of SVA transduction for 17 samples (of 35) in our dataset (transduction location for the remaining samples did not match within +/− 500 bp). We presume that this discrepancy is associated with the sample size, the accuracy of the computational methods, and the sequencing platforms used in these studies.

#### 3.2.2. Distribution of 3′ Transductions across the Populations

Our study has shown a variation in the quantity of identified transduction variants across the continental groups. We found the number of SVA transductions higher in the African super-population (AFR) than in others, whereas L1 transductions were slightly more abundant in South Asian populations (Figure 4A). In contrast, the lowest ranks belonged to the American (AMR) and European (EUR) super-populations for L1 and SVA transductions, respectively. However, these observations were not statistically significant. Regardless of the non-LTR retroelement type, our results show that the number of identified transduction variants is higher in the African super-population. Besides the sample size, this can be partly explained by the fact that the number of transductions within populations is a reflection of the number of polymorphic loci across populations. In this respect, the African super-population shows the highest number of polymorphic non-reference MEIs among others. This is consistent with Rishishwar et al.’s findings demonstrating that African populations have the highest level of polymorphic MEI genetic diversity [24]. Furthermore, these results agree with the previous studies showing higher levels of nucleotide, haplotype, structural variant, and copy number diversity in Africans compared to non-Africans [48,49,50].

We characterized the locus specificity of transduction variants to each of the five continental populations (Figure 5). It appears that many 3′ transduction loci (84.32% of L1s and 76.30% of SVAs) are only present within a specific super-population, whereas they are absent from other continental populations. Extending characterization to sub-populations, we realized that 52.98% of L1 and 28.35% of SVA transduction sites are unique to a specific population. In this regard, STU (Sri Lankan Tamil from south Asia with 10 3′ L1 transduction alleles) and GWD (Gambian in Western Divisions in the Gambia from Africa with eight 3′ SVA transduction alleles) have the highest number of population-specific alleles (Figure 6A,B). Interestingly, our results showed that one SVA-driven transduction with a size of 30 bp is present in all populations (Figure 6B). The donor locus of this transduction was detected in the RepeatMasker output of the reference genome (GRCh38 assembly) and is located within an alternate contig (chromosome 6_GL000253v2_alt:4067131-4067161), thus representing a variable genomic region. However, the offspring is a heterozygous insertion (allele count = 269) into chromosome 8, position 127613059. The source element of this specific transduction belongs to the youngest SVA family, SVA_F, which has induced several offspring in different locations (see results and discussion 3.2.3 for more details). Altogether, these observations support the recent activity of this particular locus.

#### 3.2.3. Length and Allele Frequency of 3′ Transductions

Our analyses showed that the length of transduced segments varies among different TEs and populations (Appendix A). The length of L1-driven transductions was between 8 bp and 997 bp (median = 63), whereas these ranges were 7 to 995 bp (median = 71) for SVA transductions (Table 3). Depending on the insert size of the sequencing library, MELT can only assemble a limited number of nucleotides on either side of the insert. Thus, it might have affected the maximum reported length of transduction. The count of transduction events based on their lengths and progenitors is listed in detail in Appendix A, which can be used for further inquiry. Looking at the populations, for L1s, the maximum length was observed among Africans, Americans, and Europeans, whereas the minimum size belonged to South Asian populations. As for SVA transductions, we found the maximum and minimum lengths in African and American populations. However, we could not confirm any statistically significant differences in transduction size among super-populations. 

Investigating allele count and frequency of the identified transductions showed that most of them have low allele frequency, between 0 to 0.25%, within super-populations (Figure 7A,B). In line with that, we found that 86 (of 268) L1 and 31 (of 162) SVA transductions were singleton variants, meaning there was only one allele present for the mentioned loci throughout the new panel of the 1KGP dataset. Not only a vast majority of identified transductions are present at low allele frequency across populations, but our study also indicated that 84.32% of L1 and 75.30% of SVA transductions occurred only in a solitary super-population. This feature highlights the application of non-LTR retroelements carrying transduced segments as potential genetic markers for genomic studies due to being free of homoplasy and identical by descent [24].

#### 3.2.4. Progenitors and Offspring of 3′ Transductions

Exploring the connection between the source of each transduction and corresponding offspring indicates that most transductions are interchromosomal events, in which a non-TE sequence has been retrotransposed into a locus on a different chromosome (Figure 7A,B). 

In total, we found that 58 L1s and 63 SVAs were the progenitors of all high-confidence transductions identified in this study (Table 3, Table 4 and Appendix A). Tracking down the source elements of transduced segments, we observed that 27 L1 progenitor loci had yielded more than one offspring. We located three prolific L1s; two reference progenitors—found within GRCh38—on chromosome X (coordinates: 11707248-11713279 and 11935296-11941314) and one non-reference insertion in chromosome 6 (13190801 with a length of 6017 bp) that had generated 91, 21, and 21 transductions, respectively. These master elements belong to L1HS and L1Ta, the active LINE1 elements that continue propagating across the genome. Interestingly, these numbers show that the prolific L1s within the X chromosome have produced ~42% of the total high-confidence L1 transductions. Analyzing the RepeatMasker annotation and non-reference L1 insertions across the new dataset of the 1KGP, we found that the X chromosome harbors the highest density of full-length L1s (5.64 FL L1s per 1 Mbp) among all chromosomes. This agrees with a previous study in which Bailey et al. reported that chromosome X was significantly enriched with L1 elements when compared with the autosomes [51]. Accordingly, the highly productive donor L1s identified within chromosome X might be associated with the biased L1 content observed in this chromosome. Other L1 progenitors had produced between one and nine offspring. Compared to Ebert et al., who compiled a list of active L1 elements [10], we found that ~55% of high-confidence L1 transduction sources in our study (also ~55% of low-confidence donor loci) overlapped with their list (Appendix A). Three loci on chromosome 6_GL000253v2_alt (SVA_F, 4065841-4067131), chromosome 7 (SVA_E, 20667753-20669743), chromosome 6 (SVA_F, 122847780-122849162) each with 22, 10, 9 offspring, respectively, and one within chromosome 6 (SVA_E, 56893617-56896059) with nine transductions appeared to be among the most productive SVAs. All of the aforementioned prolific SVA sources were found within the GRCh38 (i.e., reference progenitors). Other SVA sources gave rise to one to seven offspring. We also analyzed the sequence of these seven most active progenitors to identify 3′ processing-related motifs such as polyadenylation signal and upstream and downstream sequence elements as characterized by Darmon et al. [52] and Ustyantsev et al. [53]. We were able to locate a polyadenylation signal within 10–40 bp upstream of the polyadenylation cleavage site for six donors (see Table 5 for more details), which is consistent with a transduction event signature. Interestingly, considering the number of analyzed full-length SVAs for transduction discovery in comparison with full-length L1 elements (overall 4066 vs. 9847 elements), it seems that SVAs have produced more transductions (3.98% per element) than L1 elements (2.72% per element). This finding is consistent with Hoyt et al.’s work, in which they observed the same trend in the T2T-CHM13 genome [1]. Our study finds more SVA progenitors than L1 ones (63 vs. 58), suggesting SVAs might be more active, further explaining why SVAs showed a higher rate (per TE) in generating transductions. Nevertheless, other factors might affect the activity of a source element that need to be considered while interpreting the results, including the genotype zygosity status, the methylation status, and the chromatin accessibility of a full-length source element within the genome [20,54].

Notably, although the number of analyzed males in this study was almost equal to females (1599 males vs. 1603 females), we could not identify any transduction events within chromosome Y (Figure 7) (concordant with Gardner et al.’s observation [20]). Likewise, we found that chromosome Y had the fewest polymorphic MEIs among the populations; two L1s and one SVA (Figure 2B and Figure 7D). Since our transduction discovery approach is merely based on polymorphic MEIs, such a low density of poly-MEIs in chromosome Y may largely explain zero transduction incidence within this chromosome (Figure 7D). However, confounding short-read mapping caused by sequence homology between the sex chromosomes [55], the number of unknown sequences, and the repetitive nature of chromosome Y appear to have affected the discovery of MEIs and, therefore, transductions within this chromosome as well.

#### 3.2.5. Insertion Location and Functional Impact

Applying the Ensembl Variant Effect Predictor v103.0 [34] to study the insertional location and functional impact of each detected variant, we found that while many of identified TEs with transductions in this study have been inserted into intergenic and intronic regions, some have resided in functionally more critical areas such as exons of non-coding genes, UTRs, TF binding sites, and regulatory regions (Table 6 and Appendix A). Not surprisingly, the influence of all these variants on different transcript isoforms and regulatory regions was a modifier—there was no evidence of impact or predictions were difficult [34]— as the analyzed genomes were originated from healthy donors. To gain more insight into the list of genes into which transductions have occurred, we performed functional enrichment analysis using gprofiler2 [35,36], whereby eleven functional resources and biological pathways were investigated. Our analysis reveals four terms from two sources (GO biological process and GO cellular component) are significantly overrepresented (adjusted *p*-value < 0.01), which are mainly involved in neurogenesis (Figure 8). To better understand whether or not this observation is due to the influence of transductions or is a general characteristic of non-LTR retroelement insertions, we also performed the same analysis using the list of genes affected by all identified polymorphic MEI loci in this study. Consequently, besides discovering additional statistically significant GO terms and pathways, the aforementioned functional features exhibited more evidence and a larger intersection size with the annotated repositories, distinguishing them from the background. This might suggest a potential common feature among genes affected by MEIs. Although some studies on somatic tissues accentuate the role of non-LTR retroelements, particularly L1s, in the generation of diversity and complexity in the brain [56,57,58], we highly recommend conducting a comprehensive set of experiments and analyses to explore and validate our observations using the 1KGP samples (or other large genome sequencing projects).

## 4. Conclusions

As a major characteristic of non-LTR retroelement insertions, DNA transduction is a relatively common phenomenon within the human genome [1,5,6,7,20,25], which influences the genome structurally and functionally [3,4,5,7,9,13]. Our study provides a genomic catalog of putative high-confidence DNA transduction variants mediated by non-LTR retroelements (L1s and SVAs) in the new panel of the 1KGP, demonstrating the dynamic feature of these events. Given the increased sample size, improved read length and sequencing coverage of the analyzed dataset plus applied refining steps, our results appear to bring a perfect complement to the previous transduction discovery projects. We also present SVA-driven transductions that have not been identified in other studies using whole samples of the 1KGP dataset. Furthermore, we provide a complete list of low-confidence transductions (Appendix A), which can be utilized for further investigation. We put forward that the rate of 3′ SVA-driven transductions is considerable and should be identified along with L1 transductions. The telomere-to-telomere human genome consortium has recently released the first complete human genome, T2T-CHM13, which has corrected many errors and filled many gaps left in GRCh38 (such as centromeres and p-arms of acrocentric chromosomes) [60]. As a future direction, it is worth applying this genome as an alternative reference to identify transduction variants, if any, residing in these newly added regions. As a result, this will afford a full spectrum of DNA transductions that can improve our understanding of the role of non-LTR retroelements and their impact on human genome biology. In conclusion, our results signify that DNA transductions mediated by retroelements are relatively prevalent and that their identifications need to be more appreciated in genomic projects.

## Figures and Tables

**Figure 2 biology-11-01032-f002:**
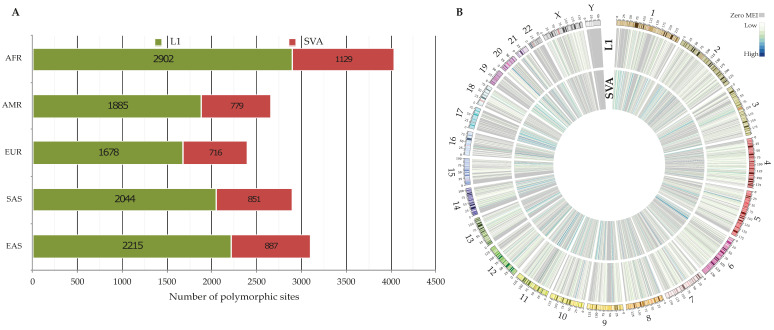
Polymorphic non-reference MEIs across 3202 genomes. (**A**) The number of identified polymorphic MEIs within super-populations. (**B**) The genome-wide density of identified MEIs in this study. The heatmaps are the number of polymorphic L1s and SVAs per 1 Mbp. The ideogram depicts the human chromosomes (GRCh38) with their cytogenetic bands.

**Figure 3 biology-11-01032-f003:**
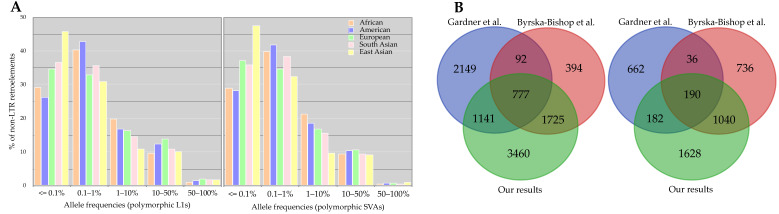
(**A**) Allele frequency distribution of polymorphic L1s and SVAs per continental population. (**B**) The comparison of discovered polymorphic loci in this study and the previous studies. The dataset used in our analysis and Byrska-Bishop et al.’s work was the same, whereas Gardner et al. studied the low-coverage samples (n = 2504).

**Figure 4 biology-11-01032-f004:**
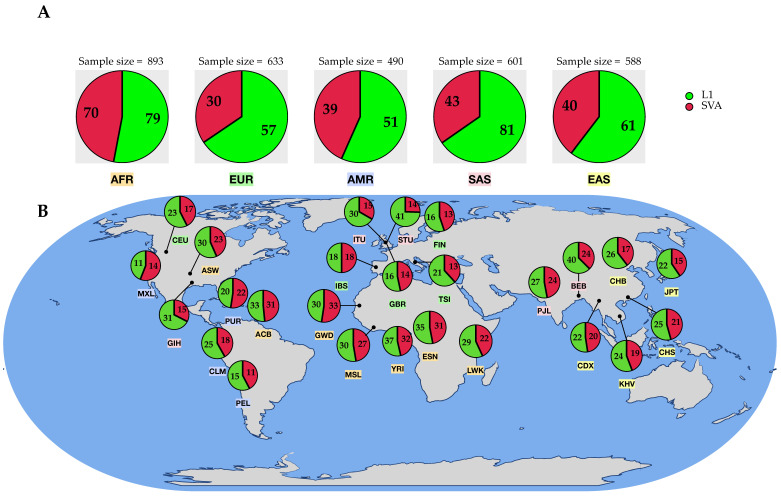
Overview of the identified (high-confidence) transduction variants based on their origin region. (**A**) Sample size and total count of identified polymorphic transductions across the five super-populations (AFR = African, AMR = American, EUR = European, SAS = South Asian, EAS = East Asian). (**B**) Number of transduction loci per sub-population. Sub-populations are color-coded based on their continental populations (Table 1).

**Figure 5 biology-11-01032-f005:**
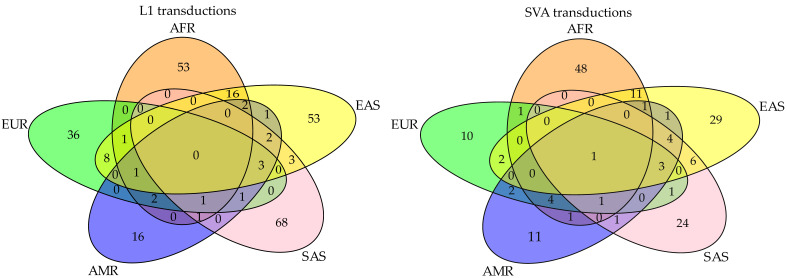
Venn diagram of high-confidence transduction loci across five super-populations. These plots represent the count and specificity of polymorphic transduction loci among the studied populations. As shown here, the majority of these loci are super-population-specific.

**Figure 6 biology-11-01032-f006:**
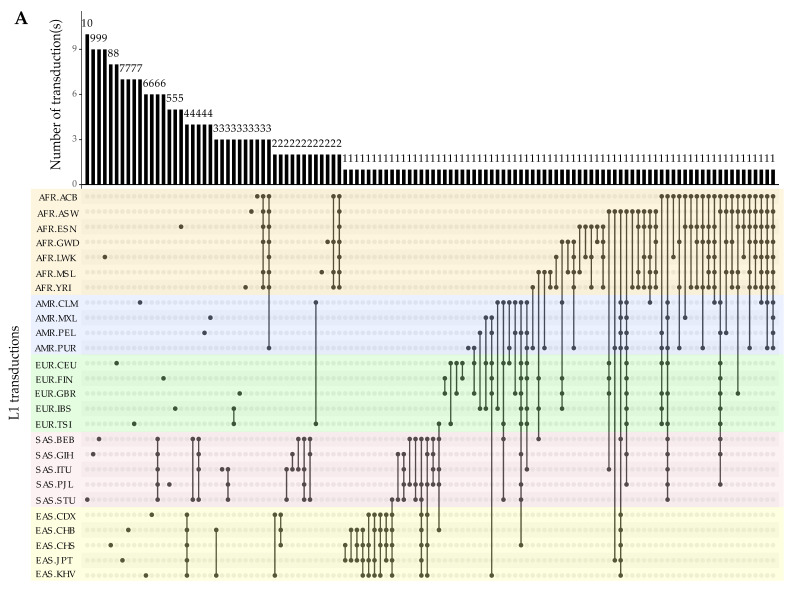
Distribution of unique and shared 3′ transduction loci (high-confidence events) across 26 populations. (**A**) Transductions mediated by L1 elements. (**B**) Transductions mediated by SVA elements. Bars above the Upset plots represent the count of either population-specific or shared loci among populations. Populations are color-coded based on their super-populations (Table 1).

**Figure 7 biology-11-01032-f007:**
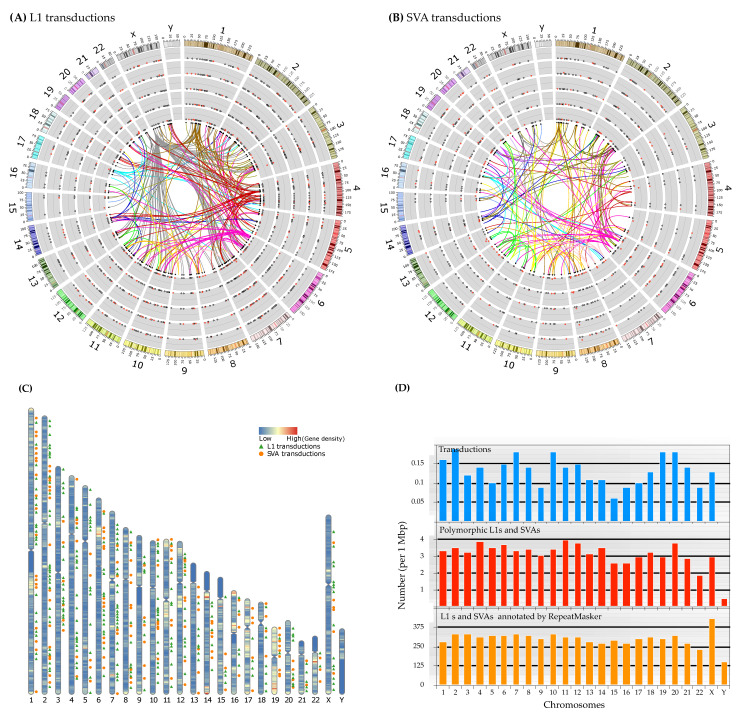
Connection(s) of progenitors and offspring, chromosomal distribution, and allele frequency of 3′ transductions across the super-populations. (**A**,**B**) The circos plots with directed links depict the genome-wide relationship of each progenitor-offspring (links are color-coded based on the chromosomal location of the source loci). Alternate haplotypes present in GRCh38 are not shown in these plots. The scatter plots represent the allele frequency of each offspring locus per super-population (from inside AFR, AMR, EUR, SAS, and EAS, respectively). Black circles within the scatter plots indicate that the allele frequency for that locus is zero. In contrast, a red dot denotes a non-zero allele frequency for a transduction locus. Allele frequencies are plotted on a scale of 0 to 0.25. In these graphs, only high-confidence events are shown. In both plots, the ideograms are the human chromosomes (GRCh38) with their cytogenetic bands. (**C**) Location and distribution of transduction events along the chromosomes with gene density heat map. (**D**) Number (per 1 Mbp) of L1s and SVAs in the GRCh38, polymorphic loci, and transduction variants compared to each other. This plot illustrates that the number of identified polymorphic MEIs and transduction variants in this study is proportional to the genome-wide L1 and SVA content.

**Figure 8 biology-11-01032-f008:**
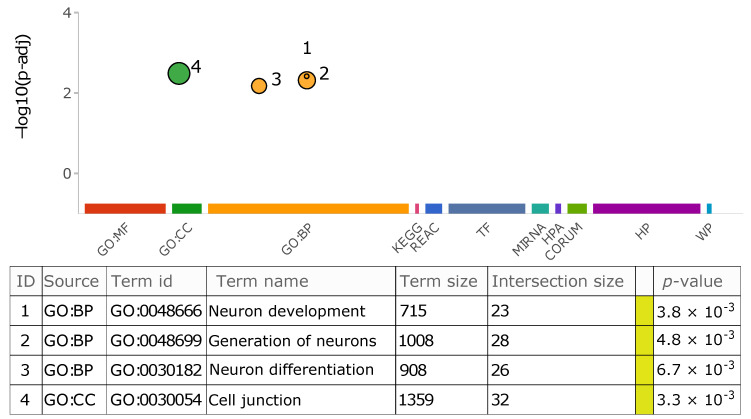
Overrepresented GO terms in the list of affected genes by transduction variants. The graph summarizes the explored repositories for functional enrichment analysis. The color-coded circles depict the significant biological features shared among those genes in which transductions have resided. The size of each circle reflects its term size. Moreover, each data point on the graph is listed in detail in the table. The followings are the plot abbreviations and definition of terms used in the table. MF: molecular function, BP: biological process, CC: cellular component, KEGG: Kyoto Encyclopedia of Genes and Genomes, REAC: Reactome database, TF: TRANSFAC database, MIRNA: miRTarBase database, HPA: Human Protein Atlas, CORUM: CORUM database, HP: Human phenotype ontology, WP: WikiPathways, Term size: number of total genes present in the corresponding repository that are annotated to the term, Intersection size: number of genes in the study list that are annotated to that term.

**Table 1 biology-11-01032-t001:** Overview of the analyzed populations from the new panel of the 1KGP (~30-fold coverage).

Super Population Code (Name)	Population Code (Name)	Sample Size
AFR (African)	ASW (Americans of African Ancestry in SW, USA)	74
	YRI (Yoruba in Ibadan, Nigeria)	178
	LWK (Luhya in Webuye, Kenya)	99
	GWD (Gambian in Western Divisions in the Gambia)	178
	MSL (Mende in Sierra Leone)	99
	ESN (Esan in Nigeria)	149
	ACB (African Caribbeans in Barbados)	116
AMR (Ad Mixed American)	MXL (Mexican Ancestry from Los Angeles, USA)	97
	PUR (Puerto Ricans from Puerto Rico)	139
	CLM (Colombians in Medellin, Colombia)	132
	PEL (Peruvians in Lima, Peru)	122
EUR (European)	CEU (Utah Residents, CEPH, with Northern and Western European Ancestry)	179
	TSI (Toscani in Italy)	107
	FIN (Finnish in Finland)	99
	GBR (British in England and Scotland)	91
	IBS (Iberian Population in Spain)	157
SAS (South Asian)	GIH (Gujarati Indian in Houston, Texas)	103
	PJL (Punjabi from Lahore, Pakistan)	146
	BEB (Bengali from Bangladesh)	131
	STU (Sri Lankan Tamil)	114
	ITU (Indian Telugu in the UK)	107
EAS (East Asian)	CHB (Han Chinese in Beijing, China)	103
	JPT (Japanese in Tokyo, Japan)	104
	CHS (Southern Han Chinese)	163
	CDX (Chinese Dai in Xishuangbanna, China)	93
	KHV (Kinh in Ho Chi Minh City, Vietnam)	122

This table summarizes the analyzed populations and their sample size. In total, 3202 whole-genome sequence data from 26 populations organized into 5 super-populations were studied. Each continental group is assigned an arbitrary color.

**Table 2 biology-11-01032-t002:** Overview of identified polymorphic non-reference MEIs in the new panel of the 1KGP.

	L1	SVA
Number of polymorphic sites	7103 *	3040 *
Minimum length of MEI (bp)	33	31
Maximum length of MEI (bp)	6019	1316
Mean length of MEI (bp)	2980	949
Median length of MEI (bp)	1975	1240

* Of these, 320 L1s and 85 SVAs are elements tagged as rSD (see Section 2.2) in the FILTER column of VCFs. According to the recommendation of MELT’s developers, these insertions were included in the analyses to increase the transduction discovery sensitivity.

**Table 3 biology-11-01032-t003:** Overview of identified 3′ transduction variants.

	L1	SVA
Total number of identified transductions (unfiltered)	505	361
Number of transductions after removing those overlapping with segmental duplicates	466	342
Number of low-confidence transductions	198	180
Number of high-confidence transductions	268	162
Number of progenitors	58	63
Minimum length of transduced sequences (bp) *	8	7
Maximum length of transduced sequences (bp) *	997	995
Mean length of transduced sequences (bp) *	206.4	205.9
Median length of transduced sequences (bp) *	63	71

* Transduction size statistics are shown only for high-confidence events.

**Table 4 biology-11-01032-t004:** Number of transduction sources based on their sub-family type.

Retroelement	Source Type	Sub-Family Type	Number
L1	Present in GRCh38	L1HS	217
	Non-reference	Sub-family undetermined	32
		L1Ta	3
		L1T1d	16
SVA	Present in GRCh38	SVA_F	83
		SVA_E	71
	Non-reference	SVA ^1^	8

A non-reference source denotes a polymorphic MEI (see Section 2.3) that has given rise to transduction. ^1^ No annotation on the sub-family type was provided by MELT for non-reference SVAs.

**Table 5 biology-11-01032-t005:** Structural features of the 3′ end of the most prolific identified donor loci in this study.

Source Locus	Source Element (Source Type)	Polyadenylation Signal ^2^	Upstream Sequence Element ^3^	Downstream Sequence Element ^4^
chrX:11707248-11713279	L1HS (reference)	AAUAAA	UGUAN	No
chrX:11935296-11941314	L1HS (reference)	AAUAAA	UGUAN	No
chr6:13190801 ^1^	L1ta1d (non-reference)	AAUAAA	-	GU reach
chr6_GL000253v2_alt :4065841-4067131	SVA_F (reference)	AUUAAA	-	GU reach
chr7:20667753-20669743	SVA_E (reference)	AAUAAA	-	GU reach
chr6:122847780-122849162	SVA_F (reference)	-	-	GU reach
chr6:56893617-56896059	SVA_E (reference)	AAUAAA	-	GU reach

^1^ Only the insertion location for this locus is reported as the source itself is a polymorphic insertion not present in the GRCh38. ^2^ The polyadenylation signal was found within 10–40 nucleotides upstream of the cleavage site. ^3^ The motif was identified within 40–100 nucleotides upstream of the polyadenylation signal. ^4^ The motif was detected within 40–100 nucleotides downstream of the polyadenylation signal.

**Table 6 biology-11-01032-t006:** Insertional location of retroelements carrying 3′ transductions.

	L1	SVA
Intergenic insertion	82	43
Intronic insertion	156	91
Exonic insertion (coding genes)	0	1 ^1^
Exonic insertion (non-coding genes)	2	4
Regulatory region insertion	13	7
TF binding site insertion	1	0
5′ UTR insertion/3′ UTR insertion	0/1	0/1
Upstream gene/Downstream gene ^2^	7/6	10/5

The insertional location of a retroelement and its transduction into the genome have been predicted using VEP (v103.0). ^1^ The affected region is exon 14 of FANCM (Fanconi Anemia Group M Protein). FANC group consists of 13 different genes required for normal activation of the Fanconi Anemia pathway [59]. ^2^ They are defined as 2 Kb upstream and 1 Kb downstream of each transcript.

## Data Availability

Not applicable.

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
