# Peer review of "A Map of 3′ DNA Transduction Variants Mediated by Non-LTR Retroelements on 3202 Human Genomes"

_biology, 2022, doi:10.3390/biology11071032_

Round 1
Reviewer 1 Report
This is an interesting original in an impact topic. Authors perform a genome‐wide map of LINE-related (inter/intrachromosomal) 3’ transduction variants from a genomic database with high coverage in the context of genomic evolution, regulatory región reorganization and human disease, and provide a genome-wide map of transduced elements.
The work gives a complete characterization of the 3’transductions in the different populations studied. While I cannot discuss the pipeline used for discovery, I would like to make two comments with the aim of improving the impact of the work.
1.- It is known that many repetitive elements are not free transcriptional units (as Pol III-Alus) but they are integrated in PolII transcripts. How is the distribution of L1/SVAs transduced elements when considering these two groups of sequences?
2.- Transduction events and retrogression are somehow related to chromatin accessibility. Could the authors relate transductions to known regions of open chromatin (e.g. DNAaseI hypersensitive regions)?
Author Response
See the file

Reviewer 2 Report
The manuscript by Reza Halabian and Wojciech Makałowski addresses an important process of genomic DNA modification by 3' DNA transduction via retroposition of non-LTR retroelements (L1 and SVA). The authors analyzed 3202 human genomes sequenced at high coverage. They characterized possible transduction-related variations and identified specific loci that are considered progenitors for a number of transduced sequences. Results are clearly presented and provide useful information about 3' transduction among a wide range of individual genomes. Some questions should be addressed before the manuscript can be accepted for publication.
1. It can be worth mentioning that the majority of L1 copies are severely 5'-truncated, which may be important for understanding their retropositional and transduction potential.
2. If there are any specific features in transduction-related elements, e.g., TSDs and 3'-end structure (AATAAA signals, Upstream Sequence Elements, or Downstream Sequence Elements for RNA polymerase II termination), they should be analyzed and indicated in the manuscript (at least in six most active loci with L1 or SVA found by the authors).
3. Progenitor elements mentioned in line 410 are very short (judging by coordinates provided in the text). Apparently, they cannot be full-length L1s capable of transposition. The USCS genome browser displays are large (close to full-length) L1 elements at these locations. Please clarify the coordinates and lengths of progenitor L1 loci.
4. Do independent offspring of the same L1 or SVA progenitor have identical or different lengths of transduced sequences?
Minor points.
1. Line 146, 161
Both web links seem to be broken and cannot be opened, please provide working links if possible.
2. In Table 1, “Mende in Sierra” should be “Mende in Sierra Leone.”
3. Line 163-165
It is unclear why transduced fragment should be shorter than its carrier, please clarify this in the text.
4. Line 234
Please explain what rSD is.
5. Fig 2B, Fig 7AB
Please describe the features displayed in the outer circle (bands, colors, and numbers) in the figure caption.
6. Fig 8
The fonts are hardly readable. Please fix the formatting of the text in this figure.
7. Line 407.
(Tables 2, 4, and S2). Table 2 or Table 3?
Author Response
See the file

Reviewer 3 Report
In the paper, Halabian and coworkers characterized the putative 3’ transduced segments by L1s and SVAs in the high coverage genome sequencing by the New York Genome Center and showed the various properties of those transduced segments (genomic location, lengths, allelic frequency across different populations). They found 268 and 162 high-confidence L1s and SVA 3’ transduction respectively. Given the possibility of the transduction event to have functional impacts such as tumorigenesis, the study is sufficiently thorough and warrants publication.
1) I have questions about the different methods of the identification of MEIs reported by the authors and other papers. The authors mentioned in line 265-276 that when compared to the MEIs identified by Bishop et. al., they managed to recover 74.79% of the calls. It would be great if the authors can describe what is special for the calls that do not overlap (i.e.: 25.21%)? How many of the significant L1 and SVA (268 and 162) overlap with those called by Bishop and coworkers?
2) In Figure 6, are the unique and shared 3’ transduction loci across populations derived from the from the high confident ones (aka the 268 L1s and 162 SVAs)?
3) In Figure 8, the formatting is slightly off with multiple characters not appearing properly. Can the authors clarify what it is shown in the paper? The same goes to some texts in Figure 1.
Author Response
See the file
